# Applications for Marine Resources in Cosmetics

**Jean-Baptiste Guillerme, Céline Couteau and Laurence Coiffard *** 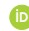

Faculty of Pharmacy, University of Nantes, Nantes Atlantique Universités, LPiC, MMS, EA2160, 9 rue Bias, F-44000 Nantes, France; jean-baptiste.guillerme1@univ-nantes.fr (J.-B.G.); celine.couteau@univ-nantes.fr (C.C.)
* Correspondence: laurence.coiffard@univ-nantes.fr; Tel.: +33-253-484-317

**Abstract:** Marine resources represent an interesting source of active ingredients for the cosmetics industry. Algae (macro and micro) are rich in proteins, amino acids, carbohydrates, vitamins (A, B, and C) and oligo-elements such as copper, iron and zinc. All those active principles play roles in hydration, firming, slimming, shine and protection. Marine organisms inhabit a wide spectrum of habitats. Photo-protective compounds can be obtained from organisms subjected to strong light radiation, such as in tropical systems or in shallow water. In the same way, molecules with antioxidant potential can be obtained from microorganisms inhabiting extreme systems such as hydrothermal vents. For example, marine bacteria collected around deep-sea hydrothermal vents produce complex and innovative polysaccharides in the laboratory which are useful in cosmetics. There are many properties that will be put forward by the cosmetic industries.

**Keywords:** algae; marine bacteria; skin; cosmetics; moisturizing; anti-aging; photo-protection

## 1. Introduction

The cosmetics industry is growing on a global scale. Since July 2013, European Commission (EC) regulation No 1223/2009 defines cosmetics as "any substance or mixture intended to be placed in contact with the external parts of the human body (epidermis, hair, nails, lips and external genital organs) or with the teeth and the mucous membranes of the oral cavity with the exclusive or principal objective to clean, perfume or protect them or, changing their appearance or keeping them in good condition". In the point of view of galenic, cosmetics are more or less complex, stable and homogeneous mixtures, resulting from a formulation which consists in the association of a raw material with another one. These ingredients are subdivided into three broad categories: active principles, excipients and additives [1].

This sector is constantly looking for innovations, especially active principles. From this point of view, the marine world is likely to open up many possibilities. Macroalgae are already widely exploited by the cosmetic industry [2], but it is still not true for microalgae and marine bacteria. However, their diversity is considerable. Only a few percentages of algae and marine bacteria have been identified and described but some are already produced on an industrial scale. They have great potential as a source of ingredients for cosmetics.

The oceans host a huge biodiversity, with more than 250,000 species described and many other species still to be discovered [3,4]. Ocean exploration in past decades allowed the discovery of a multitude of habitats sometimes in extreme environments [5]. They host a variety of organisms that produce a wide range of active molecules [6]. More than 25,000 new biologically active compounds have been identified [7]. Among marine organisms, bacteria and algae constitute a major source of active ingredients. For example, lipid content in microalgae is particularly of interest in the cosmetics domain. Some species accumulate lipids at up to 90% of their dry weight [8,9]. Microalgae are also a source of pigments, in particular carotenoids (β-carotene, lycopene, cryptoxanthin), of vitamins (A, B1, B2, B6, B12 and C) [10], canthaxantin, astaxanthin, lutein and of phycobiliproteins (phycocyanin,

phycoerythrins) [11–13]. Macroalgae represent a source of minerals, polysaccharides, proteins, lipids, and secondary metabolites such as phenolic compounds, terpenoïds, halogenated compounds, sulfur derivatives, and nitrogen derivatives that they can produce.

Bacteria are very abundant and distributed across all marine ecosystems. Many species are exploited for biotechnological applications and some would be of interest for the cosmetic industry. Indeed, many compounds from marine bacteria exhibit photo-protective, anti-aging, anti-microbial, antioxidant and moisturizing activities such as alkaloids, peptides, proteins, lipids, mycosporines and mycosporine-like amino acids, glycosides, and isoprenoids [14,15].

Coasts also hold a biodiverse array of plants including marine halophytes that still remain largely unexplored despite a high technological potential. Marine halophytes are often extremophile species equipped with potent antioxidant systems such as phenolic compounds with established beneficial therapeutic effects in humans including antioxidant and anti-inflammatory activities.

Here, we summarize some of the potential applications of macroalgae, microalgae, marine bacteria, marine fish, halophytes and marine mud and water in the fields of hydration, anti-aging, photo-protection and skin whitening.

## 2. Active Ingredients for Moisturizing Care

Maintenance of the hydration rate is essential to preserve skin integrity. Topical application of lipids or molecules that limit water loss is common. Marine organisms produce several molecules with moisturizing properties such as polysaccharides, fatty acids (sophorolipids, rhamnolipids and mannosylerythritol) and proteins that are widely used in the skin [4]. In general, the ability to restore transepidermal water loss (TEWL) to normal relies on omega 6 polyunsaturated fatty acid and specifically on the 18 carbon atoms fatty acid: linoleic acid and γ-linolenic acid [16,17]. Oil/water emulsions are formulated to avoid excessive water loss through occlusive ingredients that retain the water in the skin. Although the extracts of *Laminaria* are the choice ingredient in this case, a large number of algae can be used for this purpose. Notably, microalgae of the genus *Nannochloropsis* are of particular interest due to their high content in linolenic acid [18]. Moreover, seaweeds rich in serine, such as *Undaria pinnatifida* and microalgae of the genus *Thalassiosira* are also of particular interest [19].

Marine fish proteins mainly consist of collagen, which has been widely utilized in cosmetics for its moisturizing properties. Skin-hydrating and skin-firming of cosmetics formulated with fish derived collagen has been evaluated. Serum formulations provide an excellent moisturizing effect for a short period, whereas the cream is active when regularly applied. The cream formulations appeared to become more active later, particularly following repetitive applications [20,21]. Low doses of collagen hydrolysates derived from jellyfish have also demonstrated their potential as moisturizing agents. They increase skin hydration and reduce TEWL [22].

Ectoine (1,4,5,6-tetrahydro-2-methyl-4-pyrimidinecarboxylic acid) is an osmoprotectant produced by several bacterial species in response to osmotic stress [23]. It has been isolated for the first time from *Ectothiorhodospira halochloris* [24]. Ectoine is also produced by other halophilic bacteria, such as *alpha-* and *gamma-proteobacteria* and *Actinobacteridae* under high salt concentrations [25]. Ectoine presents a similar capacity to bind water molecules than other osmoprotectants such as glycerol [26,27]. In fact, ectoine has strong hydration properties and topical application of ectoine formulated products is well tolerated by humans [28]. Ectoine is an effective long-term moisturizer that prevents dehydration of the epidermis [23,29]. Ectoine also improves skin inflammation and is currently being investigated for the treatment of moderate atopic dermatitis [28]. Topical treatment with ectoine (EHK02-01) may represent a novel option for the treatment of patients suffering atopic dermatitis [28].

## 3. Active Ingredients to Prevent Skin Aging

Skin aging is tightly linked to extracellular matrix degradation in both epidermal and dermal layers. Intrinsic factors (genetic) are dominant; however, environmental factors also play an important

role. Among the latter ones, ultraviolet (UV) exposure, whether natural or in a tan cabin, smoking, and weather (wind exposure, for example) are important factors.

Carotenoids are major active principles among ingredients with anti-aging properties. Carotenoids are yellow/orange liposoluble pigments derived from isoprene molecules and composed of eight units of carbon atoms in which single and double bonds alternate. β-carotene tops this pigment family and has an excellent capacity to prevent reactive oxygen species (ROS) formation [30]. β-carotene is the main carotenoid produced by the halotolerant microalga *Dunaliella salina* which is able to produce more than 10% of β-carotene compare to its dry weight [31]. β-carotene is also used in anti-aging care formulations as provitamin A.

Astaxanthin applications in anti-aging care also rely on its remarkable antioxidant property [32,33], which is better than the α-tocopherol one [34]. *Haematococcus pluvialis* is the richest source of natural astaxanthin (it can accumulate more than 3 g of astaxanthin $kg^{-1}$ dry biomass) and is now cultivated at industrial scale [35–37]. Two rare carotenoids with relevant antioxidant action (i.e., saproxanthin and myxol) have been isolated from new strains of marine bacteria belonging to the family *Flavobacteriaceae* [38]. However, more investigations are required before their use in cosmetic formulations.

Among the bioactive substances with anti-aging action of marine origin, bacterial polysaccharides (PSs) are one of the most used. PSs are also produced by microalgae. In recent years, there has been a growing interest in isolating bacteria from extreme environments such as deep-sea hydrothermal vents [39–41]. It has been demonstrated that PS have properties including emulsifying, thickening, absorption and gel formation [4]. Deepsane, a PS derived from marine bacterium *Alteromonas macleodii*, has already found application in cosmetics and is commercially available [42] under the name of Abyssine® for soothing and reducing irritation of sensitive skin against chemical, mechanical and UVB aggression [40]. Anti-aging products have also been formulated with a mixture of PSs derived from *Pseudoalteromonas* sp., *Pseudoalteromonas antarctica* and *Halomonas eurihalina* that proliferate in Antarctic waters. This mixture improves skin structural properties through increased collagen I synthesis [42]. HE 800, an exo-saccharide analogous to hyaluronic acid, produced by the deep-sea bacterium *Vibrio diabolicus*, has the ability to stimulate collagen structuring [4,41].

Marine fish-derived collagen is widely used in cosmetic formulations due to its excellent skin repair and regeneration properties. Despite its origin, the marine fish-derived collagen has low odor and improved product mechanical strength. It also possesses a better absorbing capacity than collagen obtained from other animal sources [20,21].

Alguronic acid based formulations of the Algenist product range from the Solazyme Company are a mixture of polyssacharides produced by a microalga. Alguronic acid would have shown the ability to stimulate cell renewal and promote elastin synthesis [43]. Alguard® PF (Frutarom), a polysaccharide extracted from *Porphyridium sp* and proposed in the treatment of fine lines, is probably neighboring.

A *Chlorella vulgaris* extract also appears promising in the anti-age field insofar as it favors collagen synthesis [44], one of the dermis extracellular matrix macromolecules which diminishes over time resulting in wrinkle onset. A combination of algae extracts from *Meristotheca dakarensis* and *J. rubens* is available on the market which has been described as stimulating keratin, glycosaminoglycans (GAGs), and collagens I and III synthesis [2].

Hyaluronic acid is a major component of the skin extracellular matrix [45]. Inducers of hyaluronic acid synthesis are commonly used in anti-aging care. An aqueous extract of the brown alga *Macrocystis pyrifera* that belong to the *Laminariaceae* family is available on the market to that purpose. *M. pyrifera* extract may also stimulate the synthesis of syndecan-4, another important protein of the extracellular matrix [2].

The aging process reduces skin thickness, elasticity of the skin and curling of elastic fibers in the skin and gives rise to wrinkles in the skin [46]. Inhibitors of matrix metalloproteinase (MMPs) may have potential utility as an anti-wrinkle cosmetic product [47]. Matrix metalloproteinase are $Zn^{2+}$ extracellular endopeptidases enzymes produced by a variety of cells, including fibroblasts,

keratinocytes, mast cells, macrophages, and neutrophils. Three major functional groups of MMPs are described, including interstitial collagenases (degradation of type I, II, and III collagen), stromelysins (degradation of laminin, fibronectin, and proteoglycans), and gelatinases (degradation of type IV and V collagens) [48]. They play a major role in wrinkle formation [49,50]. Some studies have shown that natural and photo aging processes were linked to an increase of MMPs synthesis in fibroblast. Wrinkles occur following the cumulative impact of extensive collagen degradation by MMPs [49,50].

Innovative sources of MMPs inhibitors can be found in marine resources. The MMP inhibitory activity of marine fish-derived peptides has been studied. Peptides isolated from seahorses (SHP-1) have been shown to increase collagen release through collagenases 1, 3 and 13 inhibition [51,52]. Atlantic cod muscle also produces a gelatinase inhibitor similar to the human tissue inhibitor of MMP-2 (TIMP-2) [53]. The inhibition of MMP activity by marine-derived phlorotannins has also been investigated [54]. Many seaweed species have been evaluated for their MMP inhibitory capabilities [48]. *E. stolonifera* derived phenolic compounds eckol and dieckol showed strong inhibitions of MMP-1 expression [55]. Moreover, 6,6′-bieckol derived from *Ecklonia cava* have been shown to significantly downregulate the expressions of MMP-2 and -9 through the activation of the NF-κB pathway [56].

Sea water minerals are also known to have beneficial properties [57]. Sea water notably contains minerals (sodium, potassium, magnesium, calcium, sulfates, and chlorides) which are beneficial for the skin. Moreover, sea salts can notably be used in cosmetics for skin care [58]. Deep-sea water would have beneficial properties on general health and especially on skin health, with a positive impact on atopic dermatitis. The health benefits are claimed to be related to the minerals contained in the sea water and to the quality of the deep-sea water sources [59].

Sea mud contains various nutrients and minerals, and has been used in skin care and cosmetic product formulations for their beneficial effects and therapeutic properties on psoriasis and other skin-related disorders. Sea mud helps to retain water, equilibrates skin pH, promotes acne repair and prevention, and exhibits anti-aging properties [59,60].

However, sea water and sea mud can contain toxic elements that occur naturally or due to pollution and must therefore be subject to strict control. Notably, sea mud can entrap heavy metals due to clay's high cationic exchange capacity and positive or negative surface charge [61]. Therefore, metal impurities such as nickel and chrome can be present in cosmetic products containing natural ingredients. Notably, Dead Sea mud has been tested to determine if nickel and chrome residues can be detected. It has been demonstrated that chrome and nickel residues naturally occurred in Dead Sea mud at a low level. However, it has been demonstrated that those heavy metals are retained by the clay in solid particles. Therefore, the level of skin local exposure to Ni after mud application is at least seven-fold lower than the local toxicity threshold. Moreover, Dead Sea mud is used as a rinse-off product which limits the time of contact between the skin and the mud. However, consumers have to be aware of the presence of these metals through a clear product label to avoid their use by sensitized persons [62].

## 4. Active Ingredients for Topical Photoprotection

Three tissue layers constitute the skin; namely epidermis, dermis, and hypodermis that acts as a chemical and physical barrier. Skin can be damaged by various environmental factors, including chemicals, ultraviolet (UV), and pollution. Dermatoheliosis, also known as photo-aging, is due to UVA (400 nm < λ < 320 nm) and UVB (320 nm < λ < 290 nm) induced skin damage [63]. Prolonged human exposure to UV radiation may result in short-term and long-term effects on the skin [64,65]. The short-term effects are more or less positive. The major ones are represented by the beneficial effects on mood, the induction of vitamin D synthesis, and immediate skin pigmentation, and detrimental effects on skin thickening, actinic erythema and tanning. The long-term effects are all negatives and include photo-induced skin aging and photo-carcinogenesis related to ultraviolet radiation-induced immunosuppression. The severity of these long-term effects requires the use of appropriate protection during UV radiation exposure. Clothing and topical protection are part of the overall prevention

strategy. Sunscreens are categorized as a cosmetic in the European Union. They can be formulated using about twenty molecules only. It is therefore imperative to promote active research in this domain in order to bring out new molecules of interest.

Several marine organisms, notably photosynthetic organisms, produce UV-absorbing compounds such as scytonemins (cyanobacteria), mycosporines, mycosporine-like amino acids (MAAs), and carotenoids to protect themselves from UV radiation [66–69]. Moreover, despite their large contribution to marine biodiversity and biomass, the UV filters produced by microbial components have been poorly investigated. Marine organisms are therefore a major source of photo-protective compounds.

### 4.1. Mycosporine-Like Amino Acids

Mycosporine-like amino acids (MAAs) are intracellular water soluble colorless compounds found in many marine and freshwater organisms [70]. Freshwater microalga *Aphanizomenon flosaquae* is a good example which contains MAAs. MAAs are composed of a cyclohexenone or cyclohexenimine chromophore [71]. They are attached to the core through imine linkages, leading to a combination of resonating tautomer responsible for UV-absorption [72,73]. MAAs absorb UV radiation ranging from 310–362 nm and dissipate this energy in the form of heat radiation to the surrounding environment [74]. MAAs synthesis occurs in fungi, bacteria, cyanobacteria, phytoplankton and algae. The protection efficiency of MAAs against UV depends also on the location of these compounds in the cell. MAAs located in the cytoplasm provide a limited protection against UV while extracellular MAAs constitute a more effective shield [73,75].

### 4.2. Scytonemin

Located in the extracellular sheath of some cyanobacteria species, scytonemin is a UVA inducible pigment [73,76,77]. Scytonemin is able to reduce up to 90% of UV-A radiation into the cells due to its excellent absorption in this UV range [78–80]. It also absorbs in the UV-B range [81,82]. Oxidative stress related to UV-A exposure can also trigger synthesis of scytonemin [79].

## 5. Active Ingredients with Skin Whitening Properties

There is a great demand for whitening cosmetics for the care of lentigo, pregnancy mask, residual hyperpigmentation or hyperpigmentation following medicine poisoning. Tyrosinase is the key enzyme of melanin synthesis. Inhibitors of this enzyme are actively sought [83]. Numerous natural compounds from marine organisms have already been employed as tyrosinase inhibitors, although some of them (hydroquinones) had negative effects on human health [84]. In recent years, research focused on the discovery of new marine microorganisms derived skin-whitening compounds.

Among them, zeaxanthin seems to be of particular interest and can be obtained in *Nannochloropsis oculata* extract [85]. In the skin whitening area, a *Chlorella* extract proposed by the company Codif would also reduce skin pigmentation by more than 10%. 7-phloroeckol, a phlorotannin derived from *E. cava* brown seaweed, has been proposed as a skin-whitening agent through its anti-tyrosinase activity [86].

Marine bacteria have still not been extensively studied as a source of skin-whitening compounds. However, the marine bacteria *Pseudomonas* was found to produce the tyrosinase inhibitor methylene chloride, which reduced the pigmentation of human melanocytes [87]. The marine bacterium *Thalassotalea* sp. *PP2-459*, isolated from a bivalve, is also described to produce a *N*-acyl dehydrotyrosine derivatives tyrosinase inhibitor the thalassotalic acids [88]. Astaxanthin, which belongs to the carotenoids family, also presents interesting depigmentation properties. It would provide a protection for skin from age spots by reducing melanin production by 40% [89]. The wide majority of skin-whitening compounds used in cosmetics are still provided by terrestrial organisms, therefore opening new opportunities for marine skin whitening molecules research in cosmetics [4].

*Pistacia lentiscus* is a traditional medicinal halophyte plant of the Mediterranean area, distributed in saline environments. *P. lentiscus* leaves contain flavonoids, phenolic acids such as catechin, β-glucogallin, quercitrin gallate, gallic acid and epicatechin [90–92]. Gallic acid and epicatechins, catechins are responsible for the potent tyrosinase inhibition capacity of *P. lentiscus* and could therefore be effective in the treatment of hyperpigmentation [93–95].

## 6. Marine Resources as a Source of Excipients and Additives for Cosmetics

### 6.1. Preservatives

Preservative agents authorized for use in cosmetics are listed in the European Regulation (EC) 1223/2009, Annex V. They include parabens, a family of antimicrobial molecules that caused controversy about their safety [96,97]. Preservatives must be added to cosmetic products to prevent alteration and to exclude microbial contamination. In this context, it is extremely important to develop new and safe antimicrobial preservatives.

Among anti-microbial compounds of marine origin, macroalgae and microalgae extracts are promising. Studies highlight the inhibiting properties of extracts of macroalgae *Himanthalia elongata* and *Synechocystis spp.* regarding *Escherichia coli* and *Staphylococcus aureus* [98]. Extracts from microalgae *Isochrysis galbana*, *Chlorella marina*, *Nannochloropsis oculata*, *Dunaliella salina* and *Pavlova lutheri* showed some activity against bacteria such as *Pseudomonas aeruginosa* or *Klebsiella pneumoniae* [99]. However, those preliminary results are not sufficient to envisage the use of such extracts for industrial applications.

It could be noticed that chitosan also presents anti-microbial activity against bacteria, viruses and fungi [100]. It is a polysaccharide made of glucosamine and a variable number of GlcNAc residues obtained from chitin, a polymer abundant in marine arthropod exoskeletons and cell walls of fungi [101,102].

Falcarindiol, a polyacetylene, has been obtained from a chloroformic extract of the halophyte *Crithmum maritimum* leaves. Falcarindiol strongly inhibits the growth of different bacteria such as *Micrococcus luteus* and *Bacillus cereus*. Therefore, *Crithmum maritimum* could be potentially used in cosmetology as a preservative [103]. Extracts from another halophyte, *P. lentiscus* leaves and fruits, also exhibit anti-microbial activity [91].

Peptide with cationic moieties can interact with microbial pathogen membranes and therefore often exhibit antimicrobial properties. Antimicrobial peptides derived from marine organisms are currently studied as cosmetic applications, including lotions, shampoos, and moisture creams. Notably, the HAHp2-3-I fraction derived from the pepsin hydrolysate of half-fin anchovy (*Setipinna taty*) contained five cationic peptides (MLTTPPHAKYVLQW, SHAATKAPPKNGNY, PTAGVANALQHA, QLGTHSAQPVPF and VNVDERWRKL) that exhibit promising antibacterial potential. Antibacterials have also been isolated from Atlantic mackerel (*Scomber scombrus*) (SIFIQRFTT, RKSGDPLGR, AKPGDGAGSGPR and GLPGPLGPAGPK). They demonstrated a partial or a total inhibition property against Gram-positive (*Listeria innocua*) and Gram-negative (*Escherichia coli*) bacterial strains [104–107].

### 6.2. Essential Oil

*Crithmum maritimum* L. is a halophyte plant which grows on coastlines. *Crithmum maritimum* L. contains a combination a substances that gives to its essential oil its distinctive fragrance—lemony (due to p-cymene) but also slightly musty, of camphor and sandalwood (due to dillapiole) [108,109].

### 6.3. Antioxidant

Antioxidants provide protection against the pro-oxidative in human skin exposed to UV radiation. Antioxidants have a protective effect on human skin as they prevent damage caused by UV-induced ROS which attack membrane lipids, proteins, and DNA, such as superoxide anion, hydroxyl radicals, and $H_2O_2$. Notably, lipid oxidation by ROS participate to decrease the youthful appearance of skin [59].

Therefore, cosmeceutical industries used synthetic antioxidants such as butylated hydroxytoluene (BHT), butylated hydroxyanisole (BHA), tertbutyl hydroquinone (TBHQ), and propyl gallate (PG) to delay ROS-induced oxidation. However, these synthetic antioxidants must be used with caution due to potential health hazards [110,111]. Natural antioxidants such as phlorotannins, sulfated polysaccharides, fucosterol, and fucoxanthins derived from algae therefore represent a safe alternative for the cosmetics industry [59,112].

Carotenoids are organic pigments, also called tetraterpenoids, meaning that they are formed of 8 isoprene molecules and contain 40 carbon atoms. Over 750 carotenoids have been described and can be divided into two categories: xanthophylls, which contain oxygen, and carotenes, which are purely hydrocarbons [113]. They are mainly produced from fats by plants, bacteria and some fungi. Carotenoids have many applications including as colorants, food supplements and cosmetics/nutraceuticals [114]. Notably, carotenoids have antioxidant and anti-inflammatory properties that contribute to skin photo-protection through inhibition of UVA-induced ROS toxicity and enter in the formulation of many sunscreens [115].

Marine bacteria, yeast and fungi are an important source of carotenoids [116,117]. As an example, astaxanthin is produced by different bacteria such as *Paracoccus* and *Agrobacterium* and different yeast species, notably the following genera *Rhodotorula*, *Phaffia*, *Xanthophyllomyces* [4]. Although the production from yeasts and bacteria is lower compared to algae, yeasts have higher growth rates, easier cultivation conditions and can be genetically modified to increase carotenoid production rates [117,118].

Algae are also a major source of β-carotene [119]. As soon as the late 1960s, their potential as a β-carotene source has been investigated. *Dunaliella* is a unicellular green alga (*Chlorophyceae)* and belongs to the genus *Dunaliella salina*. *D. salina* production has been optimized to a commercial scale production of β-carotene as early as the late 1980s. It has been established that the best β-carotene production can be obtain upon high salinity and intense light [120].

The marine protist *Ulkenia* sp. and related species such as *Thraustochytriidae* sp. AS4-A1 are also able to produce antioxidants such as docosahexaenoic acid (DHA) and astaxanthin (3,3′-dihydroxy-β,β-carotene-4,4′-dione) [121], and carotenoids [122].

Marine halophytes also provide phenolic compounds with potent antioxidant activities [91,123]. The antioxidant capacity of a plant extract is usually closely related to its phenolic content [124]. Therefore, the high levels of phenolic compounds found in *L. salicaria* (278 mg GAE/g DW) could be responsible for its high antioxidant activity. High levels of phenolic compounds have also been linked to high antioxidant activity in other halophytes, including *Limonium wrightii* [125], *M. edule*, *L. monopetalum* and *T. gallica* [126–128], and *Salicornia ramosissima* [123]. Sea fennel (*Crithmum maritimum* L.) is also a halophyte of great interest because of its high secondary metabolite content. *C. maritimum* leaves notably contain carbohydrates (sucrose, glucose), organic acids (malate and quinate) and a phenolic compound, notably chlorogenic acid (CGA). Chlorogenic acid exhibits potent antioxidant activity. Depending on the ground where *Crithmum maritimum* L. grows, the plants can accumulate more or less CGA. Sand hill plants accumulate more CGA than those growing on cliffs [129].

Fish-derived proteins and peptides have also been investigated for their capacity to provide a protection to the skin from UV radiation [130,131]. Fish skin and jellyfish (*Rhopilema esculentum*) collagen and collagen hydrolysate demonstrated their ability to provide efficient protection against the detrimental effects of UV radiation, especially on the antioxidant system (superoxide dismutase and glutathione peroxidase). They not only provide protection against the degradation of skin lipids but also stimulate collagen synthesis, preventing photo-aging [132]. Collagen peptides and gelatin hydrolysate demonstrate an excellent capacity to prevent skin photo aging by hampering UV-induced inflammation, collagen destruction and preserving antioxidant enzymatic systems [133,134]

*6.4. Dyes*

Among pigments described in algae and cyanobacteria, phycobiliproteins are of great interest due to their fluorescent properties. It can be distinguished in different pigments displaying different features among the phycobiliprotein family. Phycoerythrin (PE) is a fluorescent red protein-pigment which absorbs light in the green light wavelength (λ = 498 nm and λ = 565 nm) and emits in the yellow light wavelength (λ = 573 nm). Phycocyanin is a blue accessory pigment to chlorophyll, known for its antioxidant and anti-radical properties [135–137]. Phycocyanin also has fluorescent properties and absorbs red-orange light wavelength (λ = 630 nm) and emits in the red light wavelength (λ = 660 nm). Currently, the red microalga *Porphyridium cruentum* and the cyanobacterium *Spirulina platensis* are the main sources of phycoerythrin and phycocyanin, respectively [31].

The diatom *Haslea ostrearia* has a characteristic extraplastidial color due to the accumulation of a water-soluble blue pigment to the cell apex: the marennine. This diatom is notably present in oyster refining tanks. The blue-green marennine pigment they produce is fixed in oyster gills and gives them their characteristic green color obtained after ripening. The exact nature of marennine is still unclear despite numerous biochemical characterization tests. However, it has been shown that marennine is neither carbohydrate nor protein. It would rather be a polyphenolic molecule [138].

## 7. Conclusions

There is no doubt that cosmetic formulations based on natural marine resource-derived ingredients is a good marketing argument, although this resource is still poorly exploited. The potential applications of natural molecules derived from the marine world promise a bright future for the cosmetics industry that is constantly looking for innovation. Therefore, a wide variety of marine natural products have received increased attention, especially those derived from micro- and macro-algae and marine bacteria. However, their potential is far from being fully exploited, especially for deep sea-inhabiting marine organisms that remain to be described.

Once the valuable species are clearly identified, it will remain to optimize the mode of production/extraction of the molecules of interest and to perform tests to ensure their effectiveness and their safety for cosmetic applications.

**Conflicts of Interest:** The authors declare no conflict of interest.

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
