# Peer review of "Applications for Marine Resources in Cosmetics"

_cosmetics, doi:10.3390/cosmetics4030035_

Round 1

Reviewer 1 Report

This review article did not provide enough updated informations for the readers.  Compared to a recent review entitled "Marine Microbial-Derived Molecules and Their Potential Use in Cosmeceutical and Cosmetic Products." published by Corinaldesi C. et al. (Mar Drugs. 2017 12;15(4). pii: E118. doi: 10.3390/md15040118.), this present manuscript is not sufficiently novel.  Most of the molecules extracted from various marine sources are not described in details and the related mechanisms or current applications are not well defined. 

Author Response

The manuscript was rewritten and references were added.

Reviewer 2 Report

Guillerme et al., authors of paper “Applications for marine resources in cosmetics”, proposed a review focused on potential application of macroalgae, microalgae and marine bacteria in the fields of hydration, antiaging, photoprotection and skin whitening.

The review seems well structured and the topic is very interesting and challenging. However the review is not omnicomprehensive and all issues are developed in a superficial way. No novel information are added and several others are not included. Moreover the English needs to be improved.

Therefore, I regret to suggest the rejection for this paper.

Author Response

(The authors gave the same response as above.)

Reviewer 3 Report

I suggest to delete all the information not relevant in the main context of the manuscript (eg the definition of cosmetic product, the extensive discussion on moisturization physiology, etc.), to implement the bibliographic research and to discuss more in details marine ingredients use in cosmetics and their efficacy. In the same way the introduction should be implemented in order to provide sufficient information on the manuscript background together with cutting edge information.

Author Response

(The authors gave the same response as above.)

Round 2

Reviewer 1 Report

The manuscript has been rewritten and is improved a lot. However, there are still many errors in the revision required to be corrected.

Line 73, …..avoid excessive water loss thanks to occlusive ingrediens…..??

Line 119, PSs are also produce(d) by..

Line 164, have shown to increased collagen

Line 170, to significantly down-regulated the

Line 171, MMP-2 and -9 through the activation….

Line 178, have been used in skin care…

Line 182, mud can contained toxic..

Line 184, due to the high cationic………..surface charge of clays

Line 204, are all negatives and included

Line 227, provide

Line 271, highlighted

Line 274, showed

Line 283, inhibited >>>inhibits

Line 309, used

Line 312-314 Please rewrite.

Line 332, belongs

Line 334, by

Line 349, maritimum L. grows,

Line 358, demonstrated

Line 364, It can be distinguished in different….

Line 366 and 369, absorbs

Author Response

The corrections were made

Reviewer 2 Report

The paper considerably improved in the revised version. In my opinion it is acceptable in the present form for publication.

Author Response

No comments

Reviewer 3 Report

the manuscript can be now accepted for publication.

Author Response

No comments